# Investigating Obstetric Violence in Ecuador: A Cross-Sectional Study Spanning the Last Several Years

**DOI:** 10.3390/healthcare12151480

**Published:** 2024-07-26

**Authors:** Martha Fors, Kirsten Falcón, Thais Brandao, Maria López, Desirée Mena-Tudela

**Affiliations:** 1Medical School, Universidad de Las Américas, Quito 170503, Ecuador; 2Facultad de Ciencias Médicas, Universidad de Las Américas, Quito 170503, Ecuador; kirstenfalcon@hotmail.com; 3Departamento de Psicología, Universidad de Las Américas, Quito 170503, Ecuador; thaisuni@gmail.com; 4Departamento de Enfermería, Universidad de Las Américas, Quito 170503, Ecuador; marialaura@alumni.unav.es; 5Departamento de Enfermería, Universidad Jaume I, 12006 Castelló de la Plana, Spain; dmena@uji.es

**Keywords:** obstetric violence, women rights, pregnancy, childbirth, health education

## Abstract

This pilot cross-sectional study was designed to determine the profile of obstetric violence in Ecuador in recent years. An online survey was conducted between March 2022 and April 2022, including women over 18 years who granted their informed consent to participate (n = 1598). We used non-probabilistic sampling to obtain our sample. Fisher’s exact test was performed to assess the association between violence and type of birth, healthcare facility, and education level. Out of the women who participated in the study, 89.2% (n = 1426) identified themselves as Mestiza. Additionally, 88.3% (n = 1411) had completed university-level education. The majority of the participants, specifically 63.6% (n = 1017), received their care in public institutions, and 98.2% (n = 1569) reported structural negligence, while 74.5% (n = 1190) reported violation of their right to information. The entire sample affirmed to have experienced violation of the right of presence. This report shows that obstetric violence is present in Ecuador in different ways and that women experience negligence and violation of their right to receive ethical healthcare during childbirth.

## 1. Introduction

Obstetric violence is a social problem that affects women’s rights to freely make decisions regarding their sexual and reproductive lives and bodies and prevents the guarantee of comprehensive health, negatively affecting their quality of life. This mistreatment and abuse suffered by women violates their human rights and was recently named and recognized by the United Nations [1].

Since its first legal conceptualization in Venezuela in 2007 [2], obstetric violence has become increasingly common in the Americas and around the world. In Ecuador, it has been recognized as one of the expressions of violence against women, and its scope has been extended to violence that occurs beyond the reproductive period, including all the improper or poor obstetric care that women suffer under the term gynecological and obstetric violence, presented in the Comprehensive Organic Law for the Prevention and Eradication of Gender-Based Violence Against Women [3].

Obstetric violence can be expressed in dehumanizing treatment, the abuse of medicalization, and the pathologization of women’s natural processes, including the loss of autonomy and the ability to decide freely about their bodies and sexuality. However, it also impacts the victims’ family members, depriving them of their rights to companionship and information [4], as well as health professionals who are institutionally forced to comply with standards that do not conform to the most up-to-date international or national scientific evidence or who are not well trained and suffer emotional consequences when witnessing these practices [5].

The rates of this phenomenon remain high globally [6]. In addition, despite all the evidence, including national data [4,7,8], obstetric violence continues to be widespread, invisible, and largely unknown to women users and health professionals [9]. It is important to measure and make visible this type of violence, and, for this purpose, we will present the latest national data from Ecuador. Our proposal is to contribute effectively to visualizing this type of violence and determining its extent in Ecuador.

There are a limited number of instruments for measuring obstetric violence [10]. Experiencias de Parto Relacionadas a Violencia Obstétrica (EPREVO) is a validated instrument used to measure obstetric violence in healthcare settings in Ecuador. This questionnaire was developed by a group of medical specialists and nurses from La Universidad de Las Américas [8]. Therefore, the aim of this study was to measure the level of obstetric violence in Ecuador according to the different subscales of EPREVO during the period of study.

## 2. Materials and Methods

### 2.1. Type of Study

This was an observational, descriptive, and cross-sectional pilot study. An online survey was conducted using a non-probabilistic sampling approach and launched through a periodical digital newspaper named GK, which is the most widely read independent newspaper in Ecuador. Its in-depth journalism covers human rights, gender, the environment, and transparency.

### 2.2. Participants

The inclusion criteria were women over 18 years with one or more deliveries or cesarian sections from 1990 until the time of the survey with command of the Spanish language and who granted their informed consent to participate. We excluded those questionnaires with empty responses. Information was collected from June 2021 to January 2022.

### 2.3. Demographic Variables

The sociodemographic variables that were considered were age, ethnic group, care setting, education level, obstetric history, prenatal care, and type of birth.

### 2.4. Measurement Instrument

Obstetric violence was assessed using EPREVO, a Spanish validated questionnaire that includes 46 items [8]. This questionnaire has three domains: 1. structural negligence, measuring the procedures carried out by health personnel, including physical, institutional, and intentional violence; 2. right to information, measuring the right of a woman to have information about all the procedures that are carried out by health personnel with her or her baby; 3. right to presence/supportive care, measuring a woman’s right to be accompanied during labor, childbirth, and postpartum and the right of both the mother and newborn to have immediate attachment after birth without complications.

The women who participated in the study answered the questionnaire regarding their experience of the birth of one of their children.

### 2.5. Statistical Analysis

Continuous variables were shown as means and standard deviations or as medians and interquartile range (IQR), while categorical variables were represented as absolute counts and percentages. Fisher’s exact test was performed to assess the association between violence and type of birth, healthcare facility, and education. A *p*-value < 0.05 was accepted for statistical significance. Statistical analysis was performed using SPSS v27.0 for Windows (SPSS Inc., Chicago, IL, USA).

## 3. Results

Of the 2866 questionnaires started, 1164 (41.6%) were eliminated because they were not fully answered. The final sample consisted of 1598 questionnaires.

### 3.1. Description of the Sample

The sample included 1598 respondents; 83.5% had a vaginal delivery, while the rest had a cesarean section (36.5%). The median number of children was 1.08, with an IQR of 0.01-1.00. The average number of pregnancies was 1.73, with an SD of 1.2, and the average number of abortions was 0.62, with an SD of 1.0.

The average age of the women was 27.3 years, with a standard deviation of 4.3 years.

Overall, 89.2% of the women self-identified as Mestizas. Most of the participants were treated in public institutions, and most of them attended antenatal check-ups. Approximately half of the women had some obstetric complications, while 63.5% of the complicated births ended in cesarean section. Table 1 shows the sociodemographic and obstetric characteristics of the women in the sample.

### 3.2. Types of Violence Experienced and Unnecessary Interventions

Most of the women reported having experienced structural negligence and violence in their right to information. The majority of the respondents affirmed that they had experienced the right to violence (Figure 1).

#### 3.2.1. Structural Negligence

In this study, 39.5% of the participants stated that they had received a vaginal examination less than every 4 h, 23.3% received an enema, and almost half were not allowed to choose their birth position or have food during labor, were not given correct pain control, or were not allowed to breastfeed during the first hour of life.

Of the women who underwent vaginal childbirth, 26.9% had their genitals shaved, 26.9% had undergone the Kristeller maneuver, half of them had their amniotic sac artificially ruptured, and 68.4% had an episiotomy (Table 2).

#### 3.2.2. Rights to Presence/Supportive Care

A quarter of the participating women stated that they felt guilty or humiliated, while 35.2% stated that they had received negative comments. Among the whole sample, 40.7% could not have their babies skin-to-skin. Of the total number of women, four out of ten were not allowed to be accompanied during their stay in the hospital. Approximately half of the participants were not allowed to room. In 62.6% of the patients, breastfeeding support was not received or was not sufficient (Table 3).

#### 3.2.3. Right to Information

For those women who needed to have some medication at admission to the hospital, approximately half did not receive any information about the causes of the administration. Regarding pain control, 64.9% received opportune information.

The rate of membrane rupture was not reported for half of the women. Informed consent for the cesarean section was missing for only a few of the women. The percentages of women who did not receive information about health status or an explanation of why the baby was taken away were 26.6 and 32.7, respectively; 41.6% of the participants reported the taking of unauthorized photos (Table 4).

### 3.3. Types of Violence According to Birth, Institution, and Educational Level

Women perceived greater structural violence in terms of the right to information when the birth had been by cesarean section, with statistically significant differences. Women cared for in public institutions perceived the right to information as more violent (*p* < 0.001). Women with a university education level perceived greater structural violence (*p* = 0.05) and violence on the right to information (*p* < 0.001) than did the rest of the women (Table 5).

## 4. Discussion

This study aimed to determine the prevalence of obstetric violence among women receiving obstetric care in both public and private healthcare settings in Ecuador. The study’s findings highlight obstetric violence as a significant concern within the country, with the entire sample reporting violations related to the right to presence and support during labor and delivery.

The percentage of women suffering from the three types of violence we studied was very high. In other studies, 41 to 79.7% of women experienced at least one form of obstetric violence [11,12,13,14].

Structural negligence, one of the domains of our questionnaire, was also very high in our study. In Poland, 81% of women suffer from this type of violence [15], which encompasses actions such as slapping, hitting, pinching, pushing, or any other act that inflicts physical pain or suffering, including incorrect medical procedures. A study conducted in the Gaza Strip also revealed a high prevalence of physical violence during labor, highlighting the vulnerability of women in conflict settings [11].

This form of abuse not only violates women’s fundamental human rights but can also lead to long-term physical and psychological consequences for both the mother and the baby [10].

Aspects included in supportive care for mothers and newborns were also violated in our study. Skin-to-skin contact (SCS) between mothers and babies immediately after birth, often within the context of rooming-in practices, has been widely studied for its benefits for both mothers and infants. SSC promotes breastfeeding initiation and duration, as highlighted by Jaafar [16]. This is likely due to the proximity facilitating the baby’s access to the breast and the release of hormones that support lactation. Furthermore, another study suggested that single-family rooms, which encourage rooming, can facilitate SSC and reduce parental stress. [17].

In this report, all the women suffered from violations of their right to stay informed during childbirth. A 2022 WHO report revealed that globally, only 42% of women felt they had a say in decisions made during labor and delivery, highlighting a significant gap between recommended practices and reality [18]. This lack of informed consent is further compounded by systemic issues such as power imbalances between healthcare providers and patients, particularly those from marginalized communities.

The World Health Organization emphasizes the unique ethical considerations surrounding childbirth, recognizing the profound interdependence between individuals during this process and the need for responsible actions by healthcare professionals [18].

These issues underscore potential shortcomings in healthcare provider training and preparedness, impacting the ability of healthcare providers to deliver respectful and ethical care to both women and newborns. Adherence to bioethical principles, including beneficence, nonmaleficence, autonomy, and justice, is paramount in maternity care. Violations of these principles, as observed in this study, are directly linked to the occurrence of obstetric violence.

While certain medical interventions are considered ethically nonnegotiable, obstetrics presents a unique challenge due to its inherent ethical complexities and legal considerations [19,20]. Despite the availability of ethical consultations, some physicians may prioritize their own judgment over the woman’s wishes, resorting to persuasive tactics to override her choices [21]. This behavior often stems from the inherent power imbalance within the patient–provider relationship [22]. As seen in the results, obstetric violence is an established problem in Ecuador since, for example, the entire sample has stated the violation of rights to presence/supportive care. In a study reported by Lanski et al., the main categories of obstetric violence reported were not accepted interventions/accepted interventions on the basis of partial information, undignified care, verbal abuse, and physical abuse discrimination, among others [23].

Further research is needed to explore the correlation between the dynamics of power and the prevalence of obstetric violence in Ecuador. Understanding this relationship is crucial for developing effective interventions that promote respectful and ethical maternity care practices.

Although the literature suggests a correlation between lower socioeconomic status and increased vulnerability to obstetric violence, the present study did not replicate these findings. This discrepancy may be attributed to the digital divide, which limits the participation of women with lower socioeconomic status, particularly those from marginalized communities such as indigenous populations or impoverished regions. This highlights a critical limitation in utilizing online data collection methods for studying sensitive topics such as obstetric violence. Women with the greatest need for support and advocacy may be systematically excluded due to unequal access to technology.

Despite the high global rate of antenatal care utilization (83%), the persistence of negative experiences during pregnancy and childbirth underscores the urgent need for improved obstetric care practices. Central to this improvement is upholding the principle of autonomy, which dictates respecting the decisions made by competent individuals [24].

The principle of autonomy has been defined as respect for decisions made by competent subjects [25], and women remain fully capable of making informed choices during labor and delivery when provided with accurate and comprehensible information. However, as this study suggests, gender bias continues to undermine women’s autonomy and decision-making power within maternity care settings. Further research is crucial to investigate the pervasive influence of gender bias on obstetric practices in Ecuador and identify strategies to mitigate its impact, ensuring equitable access to respectful and patient-centered care for all women [10].

Martin argues that common practices within obstetric care, such as poor informed consent processes, biased information, and lack of communication, violate basic bioethical principles. A key finding of this study is the consistent undermining of women’s autonomy in decision making during childbirth. This is evident in practices such as directed information and coercion, which limit women’s ability to make informed choices. This study also emphasizes the need for deeper ethical reflection within the field of obstetrics to address these issues and ensure respectful and dignified care for women [23].

The lack of preparedness among healthcare providers in identifying and addressing obstetric violence stems from inadequate formal training during undergraduate medical and nursing education [9]. Furthermore, the normalization of obstetric violence within healthcare systems and educational institutions perpetuates its occurrence. This study aims to contribute to the prevention of obstetric violence by disseminating its findings and advocating for systemic change. This includes influencing health authorities to integrate comprehensive education on obstetric violence into medical and nursing curricula across Ecuador and beyond. Future research should focus on developing and evaluating targeted interventions that enhance the visibility of obstetric violence among policymakers, practicing healthcare professionals, and health students.

Our findings align with previous research highlighting the prevalence of obstetric violence, particularly in the form of inadequate information provision, lack of informed consent, and limited opportunities for women to participate in decisions regarding their labor, delivery, and breastfeeding practices [26].

The observed association between higher education levels and increased cesarean section rates in Ecuador reflects trends observed in other regions. This pattern suggests a complex interplay between socioeconomic factors, patient empowerment, and the overmedicalization of childbirth. While concerns regarding the overmedicalization of childbirth and its impact on maternal and child health were raised as early as 1985, progress in addressing these issues remains slow [27].

Effectively addressing obstetric violence necessitates a multipronged approach encompassing educational reform, policy changes, and a paradigm shift within healthcare systems towards patient-centered, respectful, and rights-based maternity care.

## 5. Conclusions

This report shows that obstetric violence is present in Ecuador in different ways and that women experience negligence and violations of their rights to receive ethical healthcare during childbirth. The physical and psychological abuse of women during childbirth is a significant and ethical health problem, and urgent attention is needed to secure the rights of women who give birth. The required solution implies changes in more than one parameter, not only in legal regulations but also in health structure and health professionals’ mindsets, as well as changes in education programs to deliver highly scientific and humanitarian attention to Ecuador’s women.

### Limitations

This study has limitations that must be considered when properly interpreting the results. First, it is important to point out that the survey was online, which means that it was accessible only to women who had an internet connection. This may a limitation because women with a worse socioeconomic situation and women who belong to minority ethnic groups and who live in areas without internet connections were excluded from the study. This is necessary to consider because women in these situations tend to experience more obstetric violence than women who live in better socioeconomic and demographic conditions. We could not obtain prevalence rates because the non-probabilistic sample used was not representative of the population. Online surveys conducted via a link shared on social media platforms or websites do not allow us to define the population from which the study sample is selected. They also do not provide information about response rates. Recall bias, a type of information bias, arises when participants do not accurately remember past events or experiences, leading to inaccurate measurements, since they provided information on their experiences from 1990 until 2021. In online data collection, this bias is amplified due to the lack of direct interaction with researchers, who could clarify doubts or help stimulate memory. The accuracy of online responses is compromised by the reliance on individual memory, which can be affected by various factors, such as the passage of time, personal experience, and the influence of external information.

Second, most of the women were admitted to public institutions, where the attention given and the problems related to obstetric violence are different from those in private institutions. In addition, most of them had a university education, which may have led to better comprehension and consciousness of the situation.

In addition, as this was a cross-sectional study based on participants’ opinions via an online questionnaire, there may be information biases. Despite these limitations, the authors believe that the results obtained are relevant.

Consequently, when analyzing data collected online, it is crucial to recognize the limitations of recall bias and consider that the accuracy of responses cannot be fully guaranteed.

## Figures and Tables

**Figure 1 healthcare-12-01480-f001:**
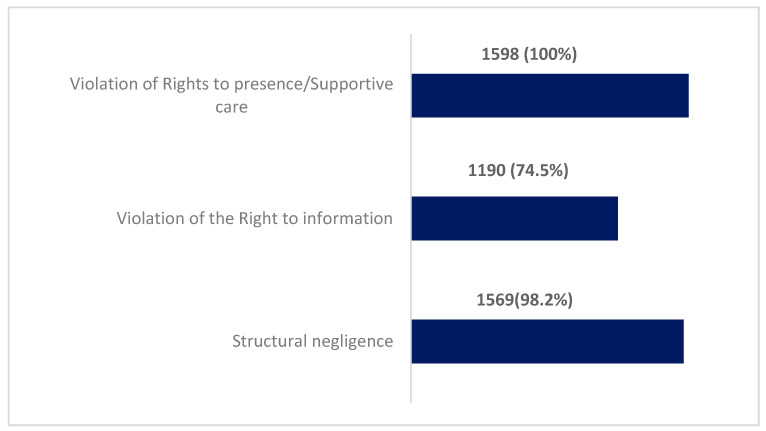
Types of detected violence (n = 1598).

**Table 1 healthcare-12-01480-t001:** Sociodemographic and obstetric characteristics (n = 1598).

Ethnicity	N	%
Indigenous	17	1.1
Black	8	0.5
Mulata	8	0.5
Montubia	33	2.1
Afro-descendant/Afro-Ecuadorian	12	0.8
Mestiza	1426	89.2
White	84	5.3
Other	10	0.6
Educational level		
Primary	16	1.0
Secondary	171	10.7
University	1411	88.3
Facility care		
Public institution	996	62.4
Private institution	233	13.9
Social security (IESS, ISPOL, ISFA)	368	23.0
Home	11	0.7
Complications during birth		
No	883	55.3
Yes	686	42.9
I do not know/I do not remember	29	1.8
Type of birth		
Cesarean section	1015	63.5
Vaginal birth	583	36.5
Antenatal classes		
No	1080	67.6
Yes	518	32.4
Antenatal care		
No	58	3.6
Yes	1540	96.4
Labor work		
No	517	32.4
Yes	1070	67.0
I do not know/I do not remember	11	0.7

**Table 2 healthcare-12-01480-t002:** Structural negligence.

	N	%
Vaginal examination less than 4 h		
No	908	56.8
Yes	631	39.5
I do not remember	59	3.7
Total	1598	100.0
Tied to the bed or stretched by the hands or feet		
No	1348	84.4
Yes	224	14.0
I do not remember	26	1.6
Total	1598	100.0
Breastfeeding during the first hour of life		
No	769	49.4
Yes	590	36.9
I do not remember	26	1.6
Not because of mother/newborn health problems	193	12.1
Total	1598	100.0
Enema		
No	1133	70.9
Yes	372	23.3
I do not remember	93	5.8
Total	1598	100.0
Kristeller maneuver		
No	394	67.6
Yes	157	26.9
I do not remember	32	5.5
Total	583 *	100.0
Genital shaving		
No	394	67.6
Yes	157	26.9
I do not remember	32	5.5
Total	583 *	100.0
Membrane rupture		
No	257	44.1
Yes	293	50.3
I do not know/I do not remember	33	5.7
Total	583 *	100.0
Episiotomy		
No	171	29.3
Yes	399	68.4
I do not know/I do not remember	13	2.2
Total	583 *	100.0

* Only women who underwent vaginal childbirth.

**Table 3 healthcare-12-01480-t003:** Right to presence.

	N	%
Humiliation/guilty		
No	1176	73.6
Yes	393	24.6
I do not know/I do not remember	29	1.8
Negative commentaries		
No	998	62.5
Yes	563	35.2
I do not know/I do not remember	37	2.3
Discrimination		
No	1376	86.1
Yes	178	11.1
I do not know/I do not remember	44	2.8
Skin to skin		
I did not have my baby skin to skin	650	40.7
Yes, less than 1 h	589	36.9
Yes, 1 h or more	125	7.8
No, due to health problems of the mother or newborn	215	13.5
I do not know/I do not remember	19	1.2
Rooming		
No	652	40.8
Yes	737	46.1
I do not know/I do not remember	13	0.8
No, because health problems of the mother or the newborn	196	12.3
Being separated from baby		
No	737	46.1
Yes	633	39.6
I do not know/I do not remember	15	0.9
Yes, because health problems of the mother or the newborn.	213	13.3
Company during the hospitalization		
No	677	42.4
Yes	844	52.8
Not all the time	77	4.8
Breastfeeding		
I did not receive support	532	33.3
It was enough	572	35.8
It was not enough	468	29.3
I do not know/I do not remember	26	1.6

**Table 4 healthcare-12-01480-t004:** Right to information.

	N	%
Information about general medication at hospital admission		
None	515	53.2
Opportune information	370	38.2
I do not know/I do not remember	73	7.6
Total	968	100.00
Information about medication for pain control		
None	262	28.9
Opportune information	587	64.9
I do not know/I do not remember	56	6.2
Total	905	100.0
Information about membrane rupture		
None	248	53.9
Opportune information	186	40.4
I do not know/I do not remember	26	5.7
Total	460	100.0
Informed consent for cesarean section		
No	175	17.2
Yes	791	77.9
I do not know/I do not remember	49	4.9
Total	1015	100.00
Authorization to feed the newborn with other milk than breast milk		
No	638	64.9
Yes	288	29.3
I do not know/I do not remember	57	5.8
Total	983	100
Information about the health status of the mother and newborn		
None	425	26.6
Opportune information	1105	69.1
I do not know/I do not remember	68	4.3
Total	1598	100.0
Explanation of why the baby was taken away		
None	277	32.7
Opportune information	530	62.7
I do not know/I do not remember	39	4.6
Total	846	100.0
Authorization for photos or videos		
No	37	41.6
Yes	48	53.9
I do not know/I do not remember	4	0.5
Total	89	100.0

**Table 5 healthcare-12-01480-t005:** Structural negligence and right to information according to kind of birth, healthcare facility, and education level.

	Structural Negligence		*p*-Value	Right to Information		*p*-Value
	No (%)	Yes (%)		No (%)	Yes (%)	
Type of birth						
Cesarean section	28 (96.6)	987 (62.9)	0.00	272 (66.7)	743 (62.4)	0.070
Vaginal childbirth	1 (3.4)	582 (37.1)		136 (33.3)	447 (37.6)	
Total	29	1569		408	1190	
Healthcare facility						
Public health facility	25 (86.2)	1350 (86.0)		336 (94.6)	989 (83.1)	
Private health facility	4 (13.8)	219 (14.0)	0.67	22 (5.4)	201 (16.9)	<0.001
Total	29	1569		408	1190	
Educational Level						
Primary	1 (3.4)	15 (1.0)		5 (1.2)	11 (0.9)	
Secondary	6 (20.7)	165 (10.5)	0.05	41 (10.0)	130 (10.9)	<0.001
University	22 (75.9)	1389 (88.5)		362 (88.7)	1049 (88.2)	
Total	29	1569		408	1190	

## Data Availability

The datasets used and/or analyzed during the current study are available from the corresponding author upon reasonable request.

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
