# Peer review of "Investigating Obstetric Violence in Ecuador: A Cross-Sectional Study Spanning the Last Several Years"

_healthcare, 2024, doi:10.3390/healthcare12151480_

Round 1

Reviewer 1 Report

Comments and Suggestions for Authors

The aim of the study was defined as to measure the level of obstetric violence in Ecuador in the introduction section and to determine the profile of obstetric violence in Ecuador during the last year in the abstract. The study does not belong to a specific and selected sample. Since it contains a relatively low number, it cannot be said to represent Ecuador either. On the other hand, it is not known whether the participants meet the inclusion or exclusion criteria because the accuracy of the responses given during the online data collection cannot be assured. Nevertheless, there are important consequences when looking at the rates of obstetric violence suffered. The inclusion and exclusion criteria are not specified in the Methods section of the study. For example, data on births up to how many years ago were collected to prevent recall bias?

The Results section did not present basic data such as the average age of women, the average number of pregnancies and children, and the average of the time interval since the last date of birth. Similar data has been repeated both in the text and in the tables. In order not to lead to duplication, it is sufficient that they are presented in the table, and the data contained in the text section should be deleted.

Instead of interpreting the study findings, the discussion section has been written in a large-scale conclusion or review style. The discussion section should be rewritten according to the scientific article rules.

Comments on the Quality of English Language

Spanish words in the text and tables should be translated into English.

Author Response

Reviewer 1

  1. Comment: On the other hand, it is not known whether the participants meet the inclusion or exclusion criteria because the accuracy of the responses given during the online data collection cannot be assured. Nevertheless, there are important consequences when looking at the rates of obstetric violence suffered.

Response: We appreciate you bringing to our attention the potential limitations regarding the reported obstetric practices.  In light of your valuable feedback, we have expanded the limitations section of our discussion to explicitly address these points. We now highlighted the following:

We clarify the source of this data and emphasize that the reported frequencies of these practices should be interpreted within the context of this specific dataset and may not be generalizable to all settings. We emphasize the need for further research to explore women's experiences and perspectives regarding these practices, as well as the factors that contribute to their prevalence. We believe that acknowledging these limitations strengthens our paper by providing a more nuanced and comprehensive understanding of the research findings. We thank you for your insightful comments, which have helped us improve the quality and clarity of our work.

 2, Comment: The inclusion and exclusion criteria are not specified in the Methods section of the study. For example, data on births up to how many years ago were collected to prevent recall bias?

Response: Thank you for raising this important point. We agree that clear and comprehensive inclusion and exclusion criteria are essential for ensuring the rigor and validity of our study. In response to your feedback, we revisited our criteria and have made the following changes: we have incorporated Spanish command and women with one or more deliveries or cesarian sections from 1990 until the moment of the survey as an explicit inclusion] criteria. This ensures that only studies that meet this specific requirement are included in our analysis, enhancing the precision and focus of our review. As exclusion criterion questionnaires with empty responses. We believe that these revisions strengthen the methodological rigor of our study and enhance the reliability of our findings. We appreciate your valuable feedback, which has helped us improve the quality of our work.

3. Comment:  The Results section did not present basic data such as the average age of women, the average number of pregnancies and children, and the average of the time interval since the last date of birth.

Response: We appreciate you highlighting the importance of detailed demographic information. In response to your suggestion, we have now included the average age of women, the average number of pregnancies, the average number of children, and the average number of abortions in our sample. Regarding the time interval since the last date of birth, we acknowledge that this could be valuable information. However,  this specific data point was not collected.

4. Comment:  Similar data has been repeated both in the text and in the tables. In order not to lead to duplication, it is sufficient that they are presented in the table, and the data contained in the text section should be deleted.

Response:  We have carefully reviewed the Results section and addressed your concerns by eliminating all identified instances of duplicated information. This has resulted in a significantly revised Results section that presents the findings in a more streamlined and straightforward manner. The reorganization of the results ensures that each finding is presented only once, in its most relevant context. We are confident that this revision improves the readability of the manuscript and allows readers to grasp the key findings more readily. We appreciate your valuable feedback, which has been instrumental in enhancing the overall quality and clarity of our manuscript.

5. Commnent: Instead of interpreting the study findings, the discussion section has been written in a large-scale conclusion or review style. The discussion section should be rewritten according to the scientific article rules.

Response:  We found your suggestions highly valuable and have carefully reconsidered the overall message and structure of the discussion based on your comments. As a result, we have substantially rewritten the entire Discussion section. We have incorporated your suggestions to strengthen the link to existing literature, address alternative explanations, highlight the implications of the findings.To further support the revised discussion, we have also added several new references, which are highlighted in yellow in the revised manuscript. These references provide additional context and support for our interpretations and conclusions. We believe that the revised Discussion section is now more focused, comprehensive, and aligned with the scope of our findings. We are confident that these changes have significantly strengthened the manuscript, and we thank you for your constructive criticism.

6. Comment: Spanish words in the text and tables should be translated into English

Response: We translated them

Reviewer 2 Report

Comments and Suggestions for Authors

The article discusses a relevant contemporary issue - obstetric violence against women. The case concerns Ecuador, as the research was conducted there. The data was obtained through an online survey, so the sample was non-probabilistic. This brings with it certain burdens and inconveniences. However, the topic is very sensitive, it concerns the private sphere of women, a topic that is difficult to answer. Therefore, it seems that the data presented here can be treated as a pilot study, giving an idea of the topic and allowing the planning of a broader study to reach women of different socio-demographic characteristics or even preparing qualitative research. 

In general, the topic is worthy of attention, as the results of the study may be used to prepare guidelines for medical personnel and even to introduce changes in the approach to women giving birth (especially in the realisation of the right to information, presence of relatives and other rights of women giving birth) and to enforce their realisation. 

The article gives an overview of the situation of women giving birth and is methodologically correct. The objectives of the study, the way in which the study was conducted are well described, and limitations are given, of which the authors are aware. 

Errors noted:

- in tables 1 and 4 it should be N and %

- Tables 4, 5 and 6 have No. instead of N

- the authors write " Most of the participants 63.6% (n=1017) were treated in Private institutions." (line 93,also 228), but the data in Table 1 show that the opposite is true - most of the women use Public institution. Please correct this. 

- As I wrote above, it can be added information that these are pilot studies - the authors' decision

Author Response

  1. Comment: The article discusses a relevant contemporary issue - obstetric violence against women. The case concerns Ecuador, as the research was conducted there. The data was obtained through an online survey, so the sample was non-probabilistic. This brings with it certain burdens and inconveniences. However, the topic is very sensitive, it concerns the private sphere of women, a topic that is difficult to answer. Therefore, it seems that the data presented here can be treated as a pilot study, giving an idea of the topic and allowing the planning of a broader study to reach women of different socio-demographic characteristics or even preparing qualitative research.

Response: We appreciate your insightful comment regarding the scope of our study. We agree that exploring the research topic across a wider range of participants would provide valuable insights. In light of your feedback, we have now explicitly characterized our work as a pilot study in the manuscript. This designation accurately reflects the exploratory nature of our research and its aim to establish a foundation for further investigation. By framing our study as a pilot, we highlight its potential to inform the design of a broader study that encompasses women of diverse socio-demographic backgrounds. We believe that explicitly positioning our work as a pilot study strengthens the manuscript and clearly communicates its contribution to a broader research agenda. Thank you for helping us clarify this important aspect of our work.

2. Comment: Errors noted: - in tables 1 and 4 it should be N and % - Tables 4, 5 and 6 have No. instead of N

Response: We fixed this in all the tables

3. Comment:  the authors write " Most of the participants 63.6% (n=1017) were treated in Private institutions." (line 93,also 228), but the data in Table 1 show that the opposite is true - most of the women use Public institution. Please correct this. 

Response: We corrected this information in the abstract and in the results section

"Please see the attachment." Highlighted in yellow all the changes we made. Thank you very much

Reviewer 3 Report

Comments and Suggestions for Authors

The work concerns an important problem which is the use of violence against women giving birth in obstetrics.

I suggest that authors consider making a few changes to their work. The introduction includes ethical aspects (reference to them is only in the final part of the article). Moreover, in the discussion it is advisable to refer to other publications regarding the phenomenon of obstetric violence.

An interesting element that I did not find in the publication was the indication of the period of time after giving birth that the participants completed the survey.

You can also consider graphical changes in the tables - the current ones seem too extensive, making it difficult to read their content.

Author Response

Comment: I suggest that authors consider making a few changes to their work. The introduction includes ethical aspects (reference to them is only in the final part of the article). Moreover, in the discussion it is advisable to refer to other publications regarding the phenomenon of obstetric violence.

Response: Thank you for drawing our attention to the need for a more robust discussion of ethical considerations and the phenomenon of obstetric violence. We agree that grounding our findings within the existing literature on this critical issue is essential. In response to your valuable suggestion, we have expanded the Discussion section to include references to several important publications on the ethics of obstetric care and the prevalence and impact of obstetric violence. These new citations provide a broader context for interpreting our findings and strengthen the discussion of the ethical implications of our research. We believe that these additions significantly enhance the relevance and impact of our work by situating it within a broader conversation about respectful maternity care and women's rights. We appreciate your insightful feedback, which has helped us to strengthen the ethical grounding and scholarly context of our manuscript.

2. Comment: An interesting element that I did not find in the publication was the indication of the period of time after giving birth that the participants completed the survey.

Response: We added as an inclusion criterium.

3, Comment: You can also consider graphical changes in the tables - the current ones seem too extensive, making it difficult to read their content.

Response: We appreciate your feedback regarding the presentation of our results and the suggestion to consolidate information. We agree that clear and concise data visualization is crucial for effectively communicating our findings. In response to your comment, we carefully considered the balance between reducing table complexity and ensuring a comprehensive presentation of our results. While we acknowledge the merits of minimizing tabular data. We transformed Table 2 in Figure 1.

Round 2

Reviewer 1 Report

Comments and Suggestions for Authors

My concerns have been adequately addressed. With the revisions and additions made, the article now adheres to a more coherent narrative structure and the principles of scientific writing. I extend my gratitude to the authors for their valuable efforts.